# Clinicopathological Significance of Pluripotent Factors in Sinonasal Intestinal-Type Adenocarcinoma

**DOI:** 10.3390/cancers17243939

**Published:** 2025-12-09

**Authors:** Federica Monaco, Alberto Vallieri, Luca Volpini, Maria P. Foschini, Alessandra Filosa, Enrica Antolini, Federico Maria Gioacchini, Giacomo Sollini, Ernesto Pasquini, Giannicola Iannella, Jiri Neuzil, Monica Amati, Lory Santarelli, Marco Tomasetti, Massimo Re

**Affiliations:** 1Department of Clinical and Molecular Sciences, Polytechnic University of Marche, 60126 Ancona, Italy; f.monaco@univpm.it (F.M.); 217454@studenti.unimore.it (A.V.); federicomaria.gioacchini@ospedaliriuniti.marche.it (F.M.G.); m.amati@univpm.it (M.A.); l.santarelli@univpm.it (L.S.); m.re@univpm.it (M.R.); 2Institute of Biotechnology, Czech Academy of Sciences, 25250 Prague, Czech Republic; luca.volpini@ibt.cas.cz (L.V.); j.neuzil@griffith.edu.au (J.N.); 3Department of Biomedical and Neuromotor Sciences, University of Bologna, 40126 Bologna, Italy; mariapia.foschini@unibo.it; 4Department of Excellence SBSP-Biomedical Sciences and Public Health, Polytechnic University of Marche, 60126 Ancona, Italy; a.filosa@univpm.it (A.F.); enrica.antolini@ospedaliriuniti.marche.it (E.A.); 5ENT Division “Bellaria Hospital”—AUSL Bologna, 40124 Bologna, Italy; giacomo.sollini@ausl.bologna.it (G.S.); ernesto.pasquini@ausl.bologna.it (E.P.); 6Department of Sensory Organs, La Sapienza University, 00185 Rome, Italy; giannicola.iannella@uniroma1.it; 7School of Pharmacy and Medical Science, Griffith University, Southport, QLD 4222, Australia

**Keywords:** sinonasal cancer, ITAC, *KLF4*, *c-MYC*, SOX2, *OCT4*, *NANOG*, prognosis

## Abstract

Sinonasal intestinal-type adenocarcinoma (ITAC) is an epithelial cancer of the nasal cavity and the paranasal sinuses, often related to work-related exposure. Because this cancer is asymptomatic, most patients with ITAC present with extensive tumors invading the surrounding tissue. In this study, genes encoding pluripotency-associated transcription factors, including *KLF4*, *c-MYC*, *SOX2*, *OCT4*, and *NANOG* (Yamanaka factors), were evaluated in malignant and non-malignant tissues of a cohort of 54 patients with ITAC, and their expressions were related to patient outcome. Our results show that a stemness phenotype with low expressions of *KFL4*, *c-MYC*, and *NANOG* and with high levels of *SOX2* and *OCT4* are predictors of a worse prognosis of ITAC. In multivariate analysis, c-MYC and *OCT4* and the type of surgery were found to be predictors of the clinical outcome. To confirm the stemness role in ITAC, ALDH1A1 expression was also evaluated. Tumor positivity to c-MYC and ALDH1A1 was associated with longer disease-specific survival, suggesting their potential role in ITAC prediction.

## 1. Introduction

Sinonasal intestinal-type adenocarcinoma (ITAC) is a rare neoplasm with overall incidence of <1% of all neoplasms, accounting for <4% of malignancies in the sinonasal area [1,2,3]. This neoplasm occurs mainly in the nasal cavity, ethmoid sinuses, and maxillary sinuses and is strongly associated with occupational exposure to wood and leather dust [4,5]. ITAC, which mimics adenoma and adenocarcinoma from the intestinal mucosa, can be divided into four histological subtypes, papillary, colonic, solid and mucinous, and mixed histology can also occur [5,6]. Due to the lack of symptoms in the early stages, this tumor is commonly detected when invading surrounding tissues. Surgical resection with endonasal endoscopic approach represents the standard-of-care treatment [7]. However, the anatomical aspects often make it difficult to delineate clear surgical margins; therefore, post-operative radiation is commonly administered considering the high recurrence rate of this tumor type; and advanced-stage tumors are also treated with neoadjuvant chemotherapy [8]. Typically, the behavior of ITAC is often characterized by relapses, recurrence, and resistance after surgical and radiation therapies. Local recurrence occurs in up to 50% of the cancer cases with intracranial invasion being the most common cause of death [9], and the 5-year overall survival rate ranges from 40 to 70% [10].

Several studies reported on the most important prognostic factors in patients with ITAC in terms of their histopathological grading, referring to the pT classification [11,12,13,14,15]. However, although the clinical staging system was developed to predict the prognosis of the disease, it is mostly based on anatomical landmarks and does not consider characteristics that are related to its biology [16]. In fact, it is not clear why in some cases ITACs present with more aggressive behavior than other cases with the same histological features and clinical stage. The most likely explanation for different behaviors occurring among apparently similar ITAC cases is related to molecular features that play an important role in a patient’s final prognosis but are still not sufficiently clarified. We believe that uncovering these prognostic biomarkers will help in advancing treatment protocols in sinonasal malignancies.

In fact, although the quality of surgical resection remains the main prognostic factor for overall survival (OS) and disease-free survival (DFS), it will be very helpful to improve our knowledge about the specific biomarkers that are related to ITAC growth and invasive capacity which can help uncover new molecular targets for therapies. Although recent research revealed molecular factors involved in the pathogenesis of ITAC and proposed them as predictors of survival [17,18,19,20,21], there is still little known about biomarkers predicting the prognosis in sinonasal ITAC, and most of these factors need to be validated [22].

Recent evidence has highlighted the role of cancer stem cells (CSCs) as an important determinant of tumor growth. CSCs have been associated with tumor relapses, metastasis, and drug resistance after standard chemo/radiotherapy [23]. The biological activities of CSCs are regulated by several pluripotent transcription factors, known as the “Yamanaka factors”, which include KLF4, c-MYC, SOX2, OCT4, and NANOG [24]. Interestingly, CSCs account for only 0.01–2% of the total tumor mass. In addition, CSCs share features with normal stem cells, including important regulatory roles of transcription factors and signaling pathways. This makes isolation and identification of CSCs challenging [25].

To evaluate the stemness role in ITAC prognosis, we analyzed, for the first time, the expressions of the Yamanaka factors *KLF4*, *c-MYC*, *SOX2*, *OCT4*, and *NANOG* in a cohort of patients with ITAC and investigate their possible roles in influencing tumor properties and survival.

## 2. Materials and Methods

### 2.1. Patients and Tissue Specimens

A total of 54 patients with sinonasal ITAC who underwent surgery were recruited at the ENT Division of “Bellaria Hospital”, AUSL Bologna, Italy, between 2011 and 2017. Inclusion criteria were sinonasal ITAC, primary surgical treatment with complete excision of the tumor obtained by endonasal endoscopic resection (ER) or endonasal endoscopic resection with transnasal craniectomy (ERTC), complete clinical data, and a minimum of three years follow-up. Patients with previous or synchronous second malignancies, previous radiation therapy or chemotherapy, or who had died of post-operative complications were excluded from the study.

Each patient was characterized on demographic and clinical bases, including age, gender, occupational exposure, site of tumor, stage, grade, subtype, type of surgery, type of margins, adjuvant therapy, DFS, OS, and the follow-up period. The follow-up period lasted till May 2022 or until the patient’s death, with a median of 48 months and survival time ranging from 2.6 to 231 months. The sinonasal ITAC stages were determined in accordance with the American Joint Committee on Cancer TNM classification of malignant tumors [14]. According to the WHO classification [15], ITACs were classified into histological subtypes indicating well–moderate–poor differentiation, typical for the papillary, colonic, and solid subtypes. The mucinous type displays mucin-filled glands or cell clusters.

All patients had undergone complete clinical examination and were staged by multiplanar computer tomography (CT), contrast-enhanced magnetic resonance imaging (MRI) or contrast-enhanced CT whenever MRI was not possible, and positron emission tomography (PET)/CT in advanced-stage lesions. Tumor tissue and its adjacent non-malignant (NM) tissue were collected and stored at −80 °C.

All patients signed a written informed consent. The study was conducted according to the Helsinki Declaration, and the samples were processed after approval of the Ethical Committee of the Marche Regional Hospital, Ancona, Italy, Rec. no. 501 of 29 November 2011.

### 2.2. Quantitative RT-PCR

Total RNA was extracted from fresh-frozen tumor and non-malignant samples (30 mg) using the RNeasy Mini Kit (item no. 74004, Qiagen, Milan, Italy) according to the manufacturer’s instructions. The concentration and purity of RNA were determined by the Nanodrop 1000 spectrophotometer (Thermo Fisher Scientific, Waltham, MA, USA). RNA integrity was assessed by Qubit™ RNA IQ Assay Kits (Thermo Fisher Scientific, Waltham, MA, USA).

The *KLF4*, *c-MYC*, *SOX2*, *OCT4*, and *NANOG* first-strand cDNA were synthesized using a High-Capacity cDNA Reverse Transcription Kit (Life Technologies, Thermo Fisher Scientific, Waltham, MA, USA). The qRT-PCR was performed using the following primers: *KLF4* (fw—CCC ACA CAG GTG AGA AAC CT; rw—ATG TGT AAG GCG AGG TGG TC); *c-MYC* (fw—ACT CTG AGG AGG AAC AAG AA; rw—GTC CAA CTT GAC CCT CTT GG); *SOX2* (fw—AGC TAC AGC ATG ATG CAG GA; rw—GGT CAT GGA GTT GTA CTG CA); *OCT4* (fw—AGC GAA CCA GTA TCG AGA AC; rw—GCC TCA AAA TCC TCT CGT TG); and *NANOG* (fw—TGA ACC TCA GCT ACA AAC AG; rw—CTG GAT GTT CTG GGT CTG GT). The genes were detected by RT-PCR using SYBR Select Master Mix (Life Technologies) in a Quant Studio 1 Real-Time PCR System (Applied Biosystem, Foster City, CA, USA). GAPDH was used as housekeeping gene (fw—TCC ACT GGC GTC TTC ACC; rw—GGC AGA GAT GAT GAC CCT TTT). Data were analyzed using the automatic cycle threshold (Ct) for the baseline and the threshold for the determination of the Ct. The samples were analyzed in duplicate, and samples with Ct values > 35 were excluded. The results were expressed as relative expression (2^−ΔCt^).

### 2.3. Immunohistochemical Analysis

Immunohistochemistry (IHC) and standard hematoxylin and eosin (H&E) staining were performed in formalin-fixed, paraffin-embedded (FFPE) tumor sections (2.5 µm). IHC was carried out by incubation with primary antibodies including anti-OCT4, anti-c-MYC IgGs, both from Dako (Camarillo, CA, USA), anti-SOX2, and anti-ADLH1A1 (Santa Cruz Biotechnology, Dallas, TX, USA). The sections were subsequently incubated with secondary antibodies (Dako, CA, USA) and visualized using the ultraView Universal DAB Detection kit (Dako, CA, USA). Nuclei were counterstained with hematoxylin. Images were acquired with an optical microscope (Zeiss, Axiocam MRc5 (Carl Zeiss Microscopy GmbH, Jena, Germany); magnification 400×). All sections and staining were assessed by an expert pathologist (GG).

### 2.4. Statistical Analysis

Results are expressed as mean ± standard deviation or as median and based on the interquartile range (IQR) and confidence interval (CI). The categorical variables were reported as fractions or percentages and compared with the chi-square method. Correlations among pluripotent factors and clinicopathological variables were performed by bivariate analysis according to the Sperman’s coefficient. Group comparisons were performed using the two-tailed Student’s *t*-test and analysis of variance (ANOVA), followed by post hoc Tukey analysis (more than two groups). Receiver operating characteristic (ROC) curve analysis was used to assess the diagnostic and prognostic sensitivity and specificity of pluripotent factors (KLF4, c-MYC, SOX2, OCT4, and NANOG), and the area under the ROC curve (AUC) was used as a diagnostic index and for prognostic accuracy. Survival analysis was applied to evaluate the cumulative probability of OS and DFS. OS was defined from the date of surgery to the time of death or to the last medical visit, while DFS was defined as the duration between the completion of treatment and the diagnosis of recurrence. According to the median value, each pluripotent factor was categorized in low (below median value) and high (above median value) expression groups. The cumulative incidence functions (CIF) of OS and DFS were estimated by the Kaplan–Meier method, and for each variable, the CIFs for different groups were compared using the log-rank test. The Cox proportional hazard model was performed in a univariate and multivariate analysis to assess the effect of prognostic factors (age, smoking, staging, histological subtype, and type of surgery) on OS and DFS. Insignificant prognostic factors were excluded from the model using backward elimination (Wald). The hazard ratios (HRs) with 95% confidence interval (CI) and *p* value were reported and visualized in the forest plots. The median follow-up was calculated according to the reverse Kaplan–Meier method. Probability values < 0.05 were considered significant. All statistical analyses were performed using the SPSS statistical package, version 25 (SPPS Inc., Chicago, IL, USA).

## 3. Results

### 3.1. Study Population

The enrolled population consisted of 54 ITAC patients, of which 93% were males, and the average age was 67 years (ranging from 33 to 92 years). Most patients were occupationally exposed to wood/leather (85%), and 44% were smokers. The tumors were in the ethmoid sinus; 3 tumors were classified as stage I, 16 as stage II, 23 as stage III, and 12 as stage IV. The ITAC included 18 papillary subtypes, 12 colonic subtypes, 15 mucinous subtypes, and 9 solid subtypes. Thirty-eight patients underwent ER and sixteen patients underwent ERTC. Intra-operative evaluation of the margins was performed, whenever possible, for radical resection. Adjuvant radiotherapy of the primary site with different techniques was delivered to 9 out of 54 patients. During the follow-up period, 22 patients (40%) developed a local relapse. The demographic and clinicopathological features are summarized in Table 1.

The study was conducted over a median (±SE) follow-up period of 42 ± 5 months [95% CI 32–52] and survival time ranging from 2.6 to 231 months. The 5-year OS was 65.3%, varying according to the histological subtype, with 80% for well-differentiated papillary subtype, 50% for the colonic subtype, 57% for the solid subtype, and 50% for the mucinous subtype.

### 3.2. KLF4, c-MYC, SOX2, OCT4, and NANOG Expression

Pluripotent factors were compared between tumor ITAC tissue and the adjacent non-malignant (NM) counterpart. *KLF4*, *SOX2*, and *NANOG* had low expressions in tumor tissue compared with its adjacent non-malignant (NM) counterpart, while no significant changes were found for *c-MYC* and *OCT4* gene expressions (Figure 1). Receiver operating characteristic (ROC) curve analysis was performed to evaluate the ability of pluripotent factors to distinguish between ITAC and NM tissue. The area under the curve (AUC) of −0.719 [0.584–0.854] for *KLF4*, −0.875 [0.774–0.975] for *SOX2*, and −0.936 [0.871–0.997] for *NANOG* significantly differentiate ITAC tissues from non-tumorous tissues (Figure 2). Bivariate analysis revealed that *SOX2* and *OCT4* positively correlated with each other and negatively with *KLF4*, *c-MYC*, and *NANOG*. Only *OCT4* correlated negatively with the tumor stage (Figure 3).

### 3.3. Survival Analysis and Association with Clinicopathological Parameters

To evaluate whether gene expressions of pluripotent factors correlate with patient prognosis, patients were divided into high and low tumor gene expression level groups by the median value (below and above the median value). Kaplan–Meier curves for high and low expression groups identified a strong relationship between the expressions of *KLF4*, *SOX2*, *OCT4*, and *NANOG* and clinical prognosis. As shown in Figure 4, patients with high expressions of *KLF4* and *NANOG* had significantly higher OS rates compared to those with low *KLF4* and *NANOG* gene expressions (log-rank, *p* = 0.049 and *p* = 0.0005, respectively). Similarly, higher *c-MYC* also showed higher OS rates, but these differences did not reach statistical significance. Conversely, patients harboring higher expressions of *SOX2* and *OCT4* exhibited lower OS rates (log-rank, *p* = 0.004). A similar scenario was observed for DFS rates in univariate analysis.

Prognostic factors, including age, smoking, histological subtypes, tumor stage, and type of surgery, were evaluated by univariate analysis. Subsequently, significant predictors were included in multivariate analysis in association with pluripotent factors.

Histological subtypes and type of surgery were significant prognostic markers predictive of poor outcome in the univariate analyses. In the multivariate survival analysis, *c-MYC* and *OCT4* levels in association with the type of surgery reached statistical significance (Table 2).

Immunohistochemistry (IHC) staining of paraffin-embedded sections was performed next to confirm the relationship between c-MYC, SOX2, and OCT4 expressions and clinical–pathological parameters. To further address the stemness and CSC potential in ITAC, ALDH1A1 expression was also evaluated. No SOX2 positivity was found in ITAC tissue, while immunoreactivity was detected for OCT4 in 4 of 16 cases (25%) and for c-MYC in 8 of 16 cases (50%). c-MYC expression was considered low (negative or weak, *n* = 8) or high (moderate or strong, *n* = 8) for further statistical analysis. Survival analysis showed no statistically significant difference in OS and DFS between high and low c-MYC-score groups (*p* = 0.744). However, patients with high c-MYC expression had better OS compared with those with low c-MYC expression as shown in Figure 5.

Tumor positivity to ALDH1A1 was found in 9 of 17 cases (52%), and its expression (cytoplasmatic or nuclear) was significantly associated with a favorable OS (*p* = 0.05) and DFS (*p* = 0.021) (Figure 6).

## 4. Discussion

The results of the present study showed, for the first time, that the expressions of KLF4, SOX2, and NANOG are significantly downregulated in ITAC samples as compared with the adjacent non-pathological tissues (while no difference was found for c-MYC and OCT4) (*p* < 0.05). The high variability in the expression of pluripotent factors found in the adjacent non-malignant counterpart may be due to the paracrine effect of tumors. Furthermore, we found that patients with tumor overexpressing KLF4 and NANOG had significantly higher OS compared to those who showed lower KLF4 and NANOG expressions (log-rank, *p* = 0.049, and *p* = 0.0005, respectively). Conversely, subjects presenting increased expressions of SOX2 and OCT4 in tumor showed lower OS rates (log-rank, *p* = 0.004).

As previously reported [26], the types of surgery and histological subtypes were significant factors associated with ITAC prognosis. Interestingly, through multivariate analysis, the type of surgery was found to be a significant predictor over the histological subtypes. Better OS was observed in patients who underwent ERTC compared to patients who had ER (*p* = 0.005). Even though ERTC is a more invasive surgery approach, providing wider access to tumor mass, better visualization, and complete removal of tumors, it may result in prolonged survival of patients. According to a multidisciplinary treatment framework, few patients received RT as adjuvant therapy (17%), which was not associated with patient outcomes. In spite of this, there is still a lack of wide literature evaluating the expression of biomarkers in ITAC of the paranasal sinuses at present, likely due to the fact that ITAC is rather infrequent.

Perez Ordonez and colleagues investigated the role of DNA mismatch repair (MMR) gene defects or disruptions of E cadherin/β catenin complex in ITAC by testing the MMR gene products, E cadherin and β catenin, in a cohort of patients with sporadic ITACs. They found that the nuclear expressions of MLH1, MSH2, MSH3, and MSH6 were preserved in these tumors, suggesting that mutations or promoter methylation of MMR genes do not play a role in ITAC pathogenesis [27]. An important result was achieved by Kennedy et al., who found that sinonasal ITACs show a distinctive phenotype, with all cases expressing CK20, CDX 2, and villin and most ITACs also expressing CK7. For this reason, it can be assumed that the expression patterns of CK7, CK20, CDX 2, and villin could be used to distinguish these tumors from other non-ITACs of the sinonasal tract [28]. We studied the status of Mir-126 and found it reduced in ITACs compared to benign tumors, suggesting the potential role of this miRNA acting as a circulating biomarker for the detection of malignant transformation [21]. Recently, we also investigated the role of MiR-let-7 in ITAC and found that its downregulation was associated with advanced-stage (pT3 and pT4) and poorly differentiated (G3) cancer (*p* < 0.05) [18].

Recently, Veuger et al. [22] systematically reviewed potential prognostic markers of ITAC. This is based on over 20 publications that report a link between biomarkers and the prognosis of sinonasal ITACs. Of the examined biomarkers, expressions of the mucin antigen sialosyl-Tn, C-erbB-2 oncoprotein, TIMP3 methylation, TP53, VEGF, ANXA2, MUC1, and the mucinous histological subtype showed a significant negative correlation with survival. Interestingly, no biomarkers were found to positively correlate with prognosis.

To explore other pathways involved in the molecular pathogenesis of ITACs, in the current study, we investigated the expression of pluripotent factors including the “Yamanaka factors” and NANOG. In our study, high KLF4 expressions directly correlated with better survival rates. Given the role of KLF4 as a negative cell cycle regulator, persistent KLF4 expression could contrast the tumor growth. KLF4 is a versatile TF involved in the regulation of numerous cellular processes, including cell growth. It was reported that KLF4 is associated with growth arrest, and its overexpression induced cell cycle arrest in several cell lines [29]. A primary mechanism by which KLF4 regulates the cell cycle includes induction of the expression of CDKN1A and inhibition of the expressions of CCND1 and CCNB1, which are involved in the progression **via** the G1/S and G2/M boundaries in the cell cycle [30]. KLF4 expression was frequently found to be lost in various human cancer types [31], and its low expression in tumors supports its tumor-suppressive function. KLF4 has emerged in a recent analysis of lung adenocarcinoma (including 497 tumors and 54 adjacent normal tissue samples) as a possible marker. Low expression of KLF4 (together with other core genes) is significantly correlated with the poor OS of lung cancer patients. Burkitt lymphoma and oral squamous cell carcinoma (OSCC) present another subtype where KLF4’s onco-suppressive role has been recently demonstrated [32]. Its function as tumor suppressor was studied on the basis of gene expression and promoter methylation approaches. IHC demonstrates that KLF4 expression decreases from well-differentiated to moderately differentiated to poorly differentiated OSCC [33]. Overall, these findings appear to be in strong agreement with our results on KLF4 expression in ITAC patients.

Nevertheless, there is evidence supporting the role of KLF4 as an oncogene. For instance, high KLF4 expression has been shown for primary breast ductal carcinoma, being associated with cell migration and invasion [34]. Moreover, an increase in KLF4 expression has been reported in human head and neck squamous cell carcinoma, and its persistent expression was associated with poor prognosis [35]. Similarly, data from our analysis show that NANOG plays an agonist role in tumor growth. High levels of NANOG expression were associated with increased malignancy, and this has been observed in many types of cancers [36], including oral squamous cell carcinoma [37].

Notably, c-MYC and OCT4 are prognostic factors in multivariate analysis. Our findings show that higher expression of *c-MYC* is related to better OS. These data are in contradiction with the current literature reporting that c-MYC expression is associated with cancer progression and metastasis of various cancers [38]. However, beyond its well-known roles in cell growth and metabolism, c-MYC significantly controls apoptosis by activating or repressing various downstream pathways [39]. A higher level of c-MYC is required for apoptosis compared to the level required to trigger cell proliferation [40]. Therefore, while deregulated c-MYC can be a potent driver of cancer growth, its role in induction apoptosis can also be viewed as a protective mechanism for the organism.

Conversely, high expression of OCT4 is associated with poor prognosis. However, our research did not confirm this, as IHC analysis revealed only four positive cases. We can postulate that there is a post-transcriptional control mechanism that may affect OCT4 protein expression in tumors. Recently, OCT4 has emerged as a master regulator that controls pluripotency, self-renewal, and maintenance of stem cells [41]. High expression of OCT4 was linked to worse prognosis in patients with solid tumors including hepatocellular carcinoma, esophageal squamous cell carcinoma, gastric cancer, cervical cancer, and colorectal cancer [42]. Also, high expressions of OCT4 were observed in breast CSC-like cells (CD44+/CD24−) [43]. There is data showing that OCT4 expression is associated with poor clinical outcome in hormone receptor-positive breast cancer [44].

The expression of OCT4 parallels the expression of SOX2, showing a positive mutual correlation. Both OCT4 and SOX2 are master pluripotent factors serving as a molecular switch that drives the fate of CSCs during cancer progression, with proven clinical potential. In fact, SOX2 expression is much higher in tumor tissues than in normal tissues, and a high level of SOX2 correlates with poor prognosis [43]. Although we found low SOX2 expression in ITAC with respect to non-malignant tissue, its high-level expression within tumor was associated with poor prognosis. Modification or abnormal expression of SOX2 has been implicated in the occurrence, progression, invasion, and metastasis of breast and lung cancers [45]. Concerning the head and neck region, the expression of SOX2 was found much higher in laryngeal carcinoma tissues, and high SOX2 expression is associated with late clinical stage and early recurrence in laryngeal carcinoma. The expression of SOX2 affects the OS of patients, acting as an independent prognostic factor for laryngeal squamous cell carcinoma tissues of patients, indicating that SOX2 may present as a useful prognostic marker and a potential therapeutic target for laryngeal squamous cell carcinoma patients [46,47]. In sinonasal cancer, SOX2 protein expression was highly heterogeneous among different histologies, with SCC showing the highest protein expression [48]. In addition, SOX2 expression in association with βIII-tubulin is common in poorly differentiated sinonasal tumors [49].

Given the role of stemness in the development and progression of ITAC, ALDH1A1, a factor involved in the regulation of gene expression in CSCs, was also investigated in IHC. Although ALDH1A1 has been associated with tumor progression in different cancers [50], we identified a tumor suppressor role of ALDH1A1 in ITAC. Weak or moderate expression level of ALDH1A1 was significantly associated with longer disease-specific survival. A dual ALDH1A1 behavior has been described. ALDH1A1 overexpression was associated with either a better or a worse prognosis, depending on its expression. While weakly stained was associated with a better prognosis, strongly stained was associated with a worse prognosis [51].

## 5. Conclusions

Despite being limited by a low sample number, single-center design, and lack of validation in independent cohorts, our results highlight that a stemness phenotype with low expressions of KFL4, c-MYC, and NANOG and high levels of SOX2 and OCT4 can be used as a predictor of worse prognosis in ITAC. In multivariate analysis, high c-MYC and low OCT4 were associated with better clinical outcome. The association between c-MYC and ALDH1A1 positivity and favorable prognosis of ITAC suggest that rather than having a role in stemness, these factors act as tumor suppressors and may be used as markers of prognosis and response to treatment in ITAC patients.

## Figures and Tables

**Figure 1 cancers-17-03939-f001:**
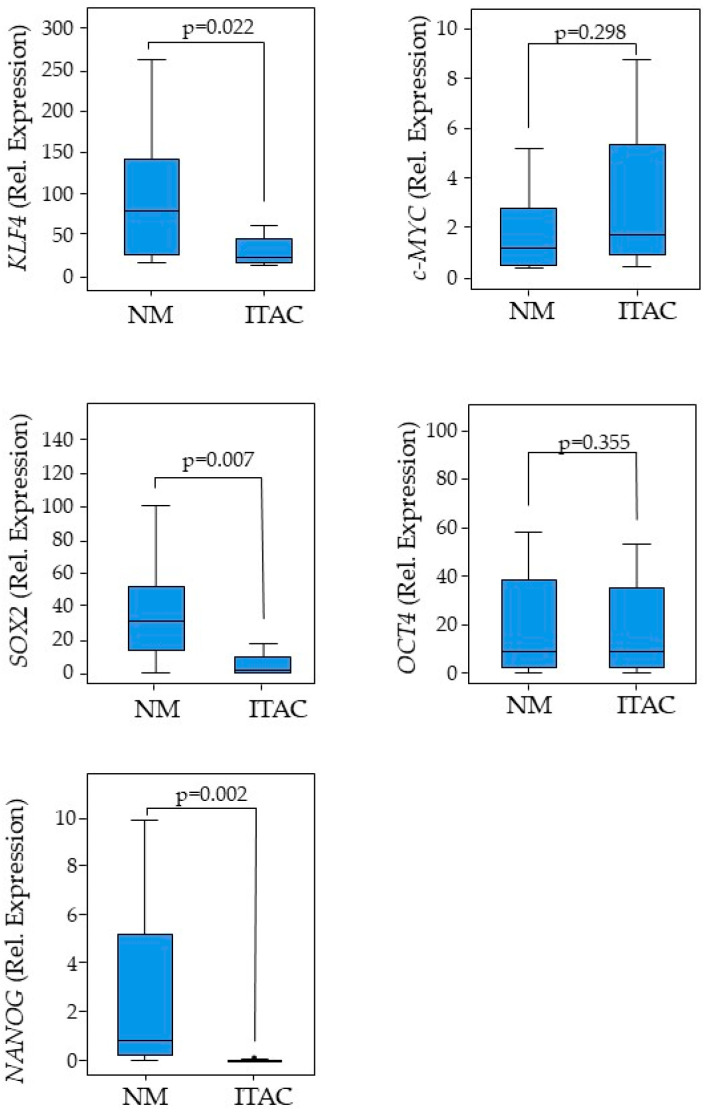
Distribution of pluripotent factors. Expressions of KLF4, c-MYC, SOX2, OCT4, and NANOG in ITAC tissues and paired adjacent non-malignant (NM) tissues. Box-plots show the median and interquartile range. Comparisons between groups were determined by *t*-test analysis. Differences with *p* < 0.05 were considered statistically significant.

**Figure 2 cancers-17-03939-f002:**
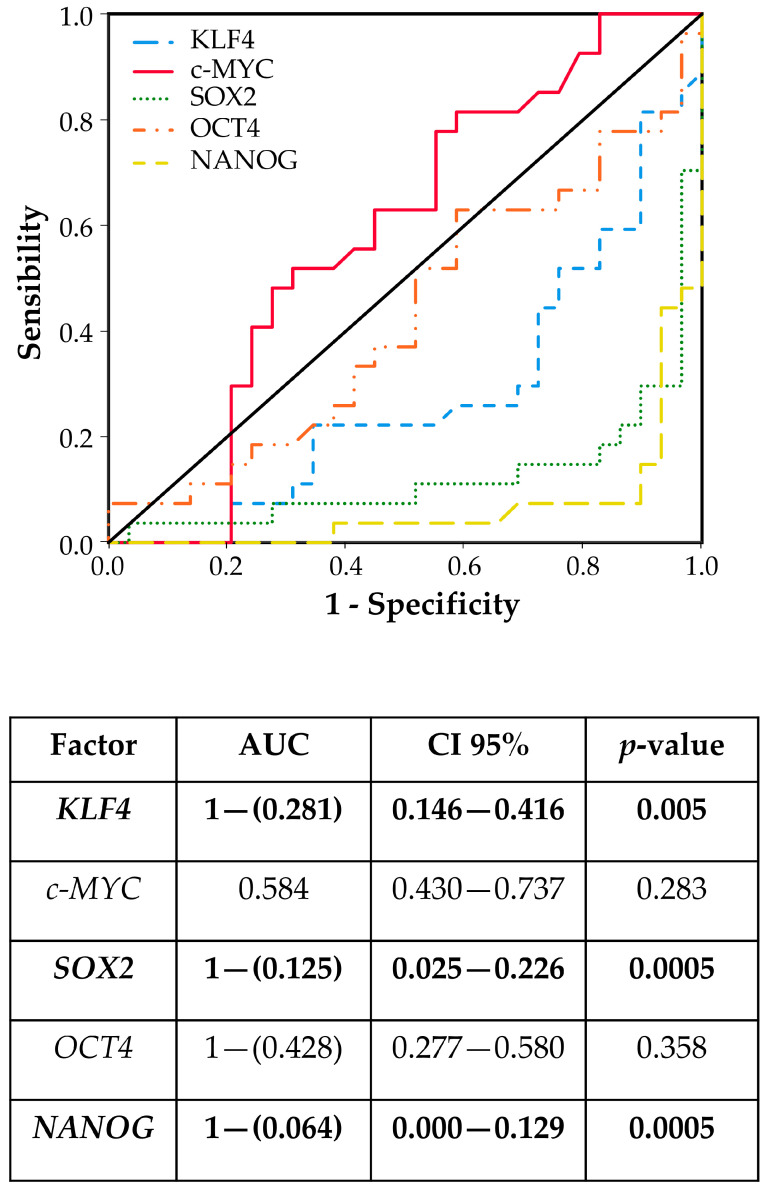
The receiver operating characteristic (ROC) models and the area under curve (AUC) with confidence interval (CI) document the sensitivity and specificity in differentiating ITAC tissues from non-malignant mucosa tissues. Differences with *p* < 0.05 were considered statistically significant (marked in bold).

**Figure 3 cancers-17-03939-f003:**
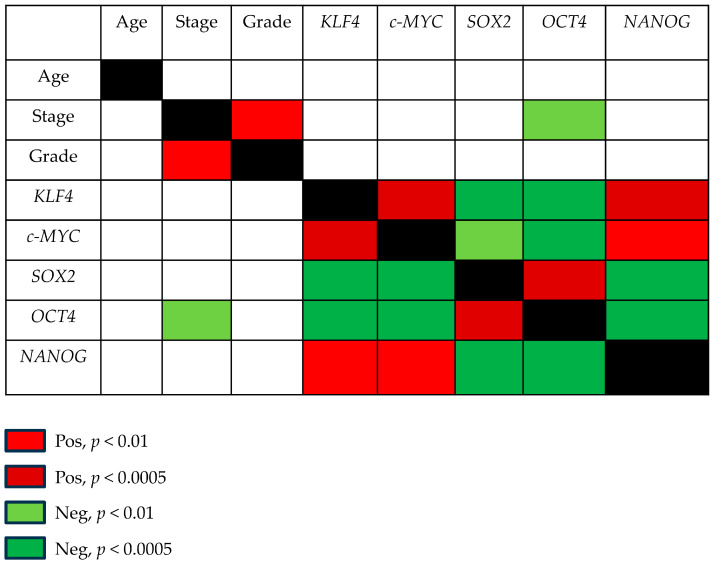
Correlations among pluripotent factors and clinic-pathological parameters in ITAC. Correlations between KLF4, c-MYC, SOX2, OCT4, and NANOG with each other and with age, tumor stage, and grade were determined by Spearman’s test. *p* < 0.05 was considered significant. Significantly positive and negative correlations were highlighted in red and green, respectively.

**Figure 4 cancers-17-03939-f004:**
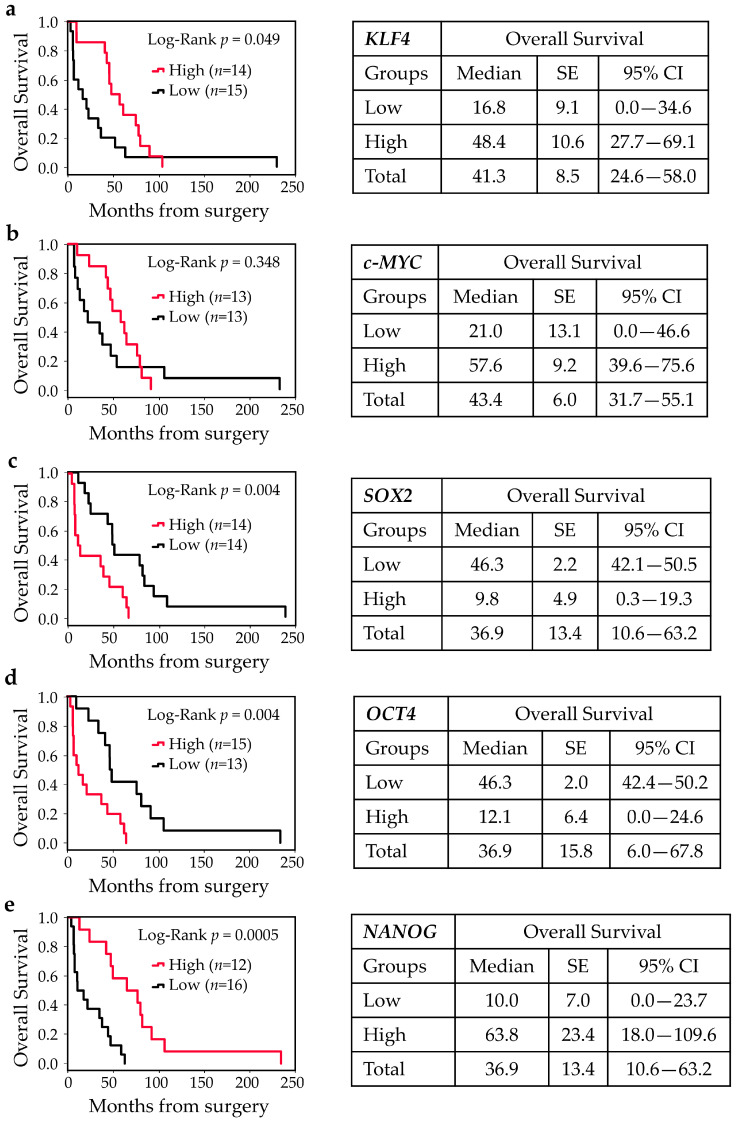
Kaplan–Meier survival curves for ITAC stratified into two groups according to the median value for KLF4, c-MYC, SOX2, OCT4, and NANOG expressions. Low and high expressions of KLF4 (**a**), c-MYC (**b**), SOX2 (**c**), OCT4 (**d**), and NANOG (**e**) were associated with overall survival (OS). Comparisons between groups were made using log-rank test, and two-sided *p* <  0.05 was considered statistically significant. Median values, standard error (SE), and a 95% confidence interval (CI) [minimum–maximum] are summarized in the chart on the right.

**Figure 5 cancers-17-03939-f005:**
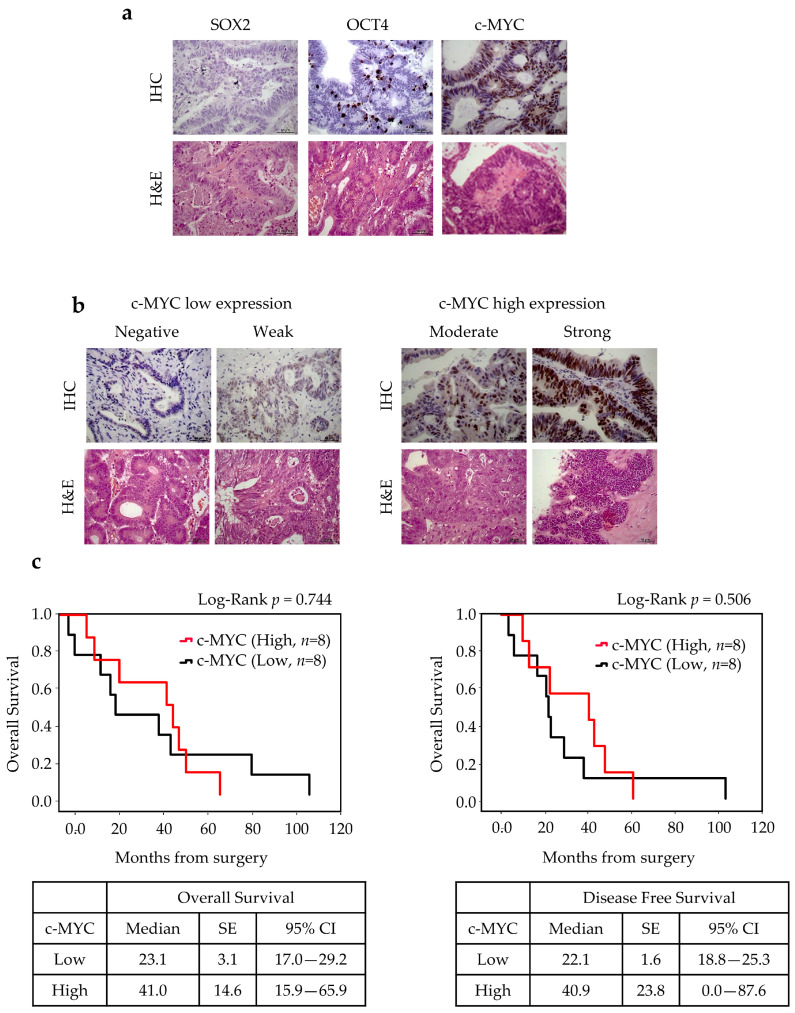
SOX2, OCT4, c-MYC expressions in ITAC tissue and survival rates. (**a**) Immunohistochemical analysis of expressions of SOX2, OCT4, and c-MYC in primary ITAC specimens (magnification 400×). (**b**) c-MYC low and high expression groups. (**c**) The OS and DFS according to c-MYC expression using the Kaplan survival curve. Comparisons between groups were made using log-rank test, and two-sided *p* <  0.05 was considered statistically significant. Median values, standard error (SE), and a 95% confidence interval (CI) [minimum–maximum] are summarized in the chart on the right. Scale bar, 50 μm.

**Figure 6 cancers-17-03939-f006:**
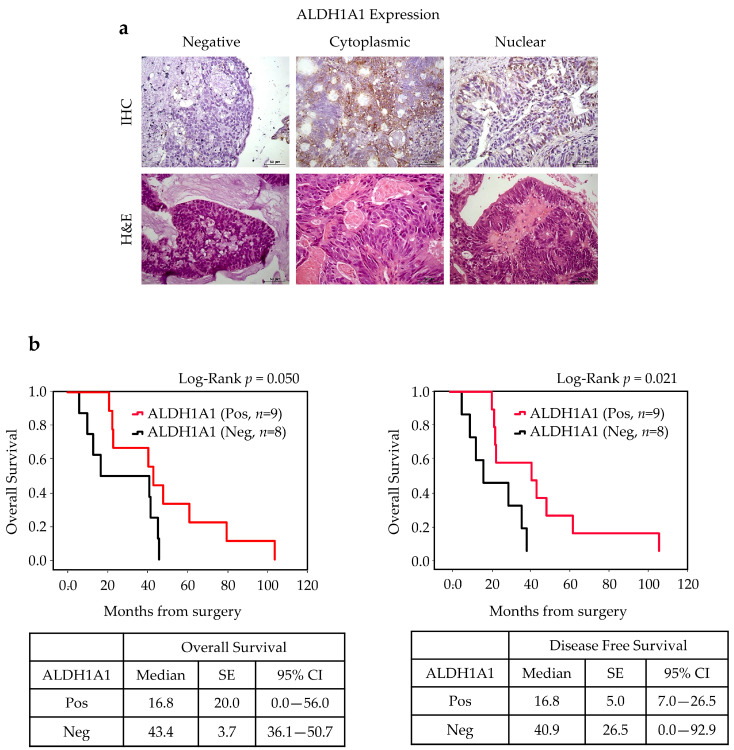
ALDH1A1 expression in ITAC tissue and survival rates. (**a**) Immunohistochemical analysis of expression of ALDH1A1 in primary ITAC specimens (magnification 400×). (**b**) The OS and DFS according to ALDH1A1 positivity using the Kaplan survival curve. Comparisons between groups were made using log-rank test, and two-sided *p* < 0.05 was considered statistically significant. Median values, standard error (SE), and a 95% confidence interval (CI) [minimum–maximum] are summarized in the chart on the right. Scale bar, 50 μm.

**Table 1 cancers-17-03939-t001:** Demographic and clinicopathological characteristics of ITAC patients.

Age (yrs ± SD)	67 ± 13
Gender (No. %)	
Male	50 (93)
Female	4 (7)
Smoking (No. %)	
Non-smoker	18 (33)
Smoker	28 (52)
Former	8 (15)
Wood/Leather exposure (No. %)	
Yes	46 (85)
No	8 (15)
Occupational task (No. %)	
Carpenter	30 (55)
Cobblers	15 (28)
Other	9 (17)
Type of Surgery (No. %)	
ER	38 (70)
ERTC	16 (30)
Adjuvant therapy (No. %)	
None	45 (83)
RT	9 (17)
Subtype (No. %)	
Papillary	18 (33)
Colonic	12 (22)
Solid	9 (17)
Mucinous	15 (28)
Grading (No. %)	
G1	4 (7)
G2	42 (78)
G3/4	8 (15)
TNM (No. %)	
S-I	3 (5)
S-II	16 (30)
S-III	23 (43)
S-IV	12 (22)
Relapse (No. %)	
No	32 (60)
Yes	22 (40)

ER, endoscopic resection; ERTC, ER–transnasal craniectomy; RT, radiotherapy.

**Table 2 cancers-17-03939-t002:** COX-regression analysis and forest plot for the subgroup analysis of the relationship between prognostic factors and OS or DFS.

**Overall Survival**	***p*-Value**	**HR (95% CI)**	** 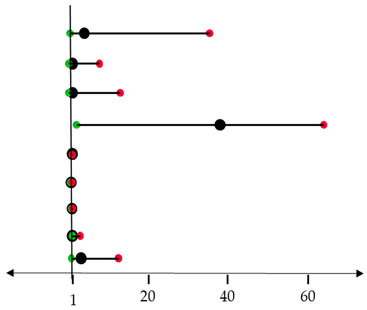 **	
**Histological subtype**	0.288			
Papillary (*n* = 16)	reference			
Colic (*n* = 8)	0.210	4.023 (0.457–35.402)		
Solid (*n* = 8)	0.914	1.113 (0.159–7.798)		
Mucinous (*n* = 13)	0.933	1.111 (0.095–13.026)		
**Type of surgery**	**0.014**	**38.016 (2.110–64.921)**		
***KLF4*** (*n* = 32)	0.235	1.019 (0.988–1.052)		
***c-MYC*** (*n* = 30)	**0.045**	**0.770 (0.597–0.994)**		
***SOX2***(*n* = 32)	0.412	0.940 (0.811–1.090)		
***OCT4*** (*n* = 32)	**0.002**	**1.004 (1.002–3.012)**		
***NANOG*** (*n* = 30)	0.088	3.257 (0.840–12.625)	Low risk	High risk
**Disease Free Survival**	***p*-**v**alue**	**HR (95% CI)**	* 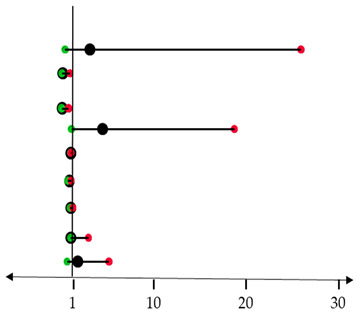 *	
**Histological subtype**	0.264			
Papillary (*n* = 16)	reference			
Colic (*n* = 8)	0.301	3.080 (0.365–26.010)		
Solid (*n* = 8)	0.438	0.105 (0.012–0.908)		
Mucinous (*n* = 13)	0.027	0.059 (0.005–0.724)		
**Type of surgery**	**0.042**	**4.455 (1.058–18.759)**		
***KLF4*** (*n* = 32)	0.917	0.999 (0.988–1.010)		
***c-MYC*** (*n* = 30)	0.098	0.834 (0.672–1.034)		
***SOX2*** (*n* = 32)	0.905	1.012 (0.834–1.227)		
***OCT4*** (*n* = 32)	**0.016**	**1.002 (1.000–2.904)**		
***NANOG*** (*n* = 30)	0.298	1.764 (0.606–5.131)	Low risk	High risk

Differences with *p* < 0.05 were considered statistically significant and marked in bold. Circles (black) represent the hazard ratio (HR) and the horizontal bars extend from the minimum (green) to the maximum (red) of the 95% confidence interval (CI) of the estimate of the hazard ratio.

## Data Availability

Data are available upon request.

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
