# Peer review of "Clinicopathological Significance of Pluripotent Factors in Sinonasal Intestinal-Type Adenocarcinoma"

_cancers, 2025, doi:10.3390/cancers17243939_

Round 1
Reviewer 1 Report
Comments and Suggestions for Authors
This is an interesting and well-structured manuscript that addresses the prognostic significance of pluripotency-associated transcription factors in ITAC, a rare and understudied malignancy. The topic is clinically relevant and potentially impactful as it explores molecular biomarkers that could inform personalised treatment strategies. The manuscript is generally well written, with clear descriptions of the methodology. However, there are several issues that should be addressed before publication. Below, I outline general and specific comments intended to improve the scientific rigour, clarity and clinical utility of the manuscript.
Major comments:
Language and grammar: There are multiple grammatical issues throughout the manuscript that affect clarity. I recommend careful proofreading by a native English speaker or a professional editing service. For example: On line 20, 'their expression has been related...' should be 'has been related'. Moreover, inconsistent formatting of gene names needs to be fixed (e.g. KLF4, SOX2). Please follow standard gene/protein nomenclature (genes in italics and proteins in capital letters without italics).
RNA quality and sample integrity: The methods section should describe how RNA integrity was assessed beyond concentration and purity. Were samples of poor quality excluded? This is crucial for reproducibility and data robustness.
Cut-off: The manuscript refers to 'high' and 'low' expression groups in the Kaplan–Meier analysis, but it is unclear how these were defined. Please clarify which statistical methods were used to determine the cut-offs.
Lack of protein-level validation: all findings are based on mRNA levels. Given the complexity of post-transcriptional regulation, the absence of protein validation (e.g. by immunohistochemistry) undermines the translational value of the findings. While not mandatory, a brief discussion of this limitation is necessary.
Survival and Prognostic Analysis: SOX2 is reported as an independent prognostic factor. Please clarify whether the multivariate analysis included all clinical variables (e.g. age, stage, smoking status and subtype) and specify the number of events per variable, to ensure the model is valid. Consider adding a forest plot to visualise hazard ratios.
Contradictory interpretation of c-MYC: although higher c-MYC expression appears to be associated with better survival, the manuscript acknowledges that this is not statistically significant. This point is overemphasised in the discussion. It should be revised and toned down accordingly.
The discussion would benefit from more detailed elaboration on the clinical implications and applications of these findings, especially the potential utility of SOX2 as a prognostic marker. Is there potential for risk stratification or treatment tailoring?
Limitations: Please include a dedicated paragraph discussing the limitations of the study, such as the small sample size, single-centre design, lack of validation in independent cohorts and no protein-level confirmation.
Minor Comments:
Figures: Improve the contrast and font sizes of all figures. Legends should be fully self-explanatory.
Terminology: Consistently define and abbreviate “endonasal endoscopic resection” (ER) and “endonasal endoscopic resection with transnasal craniectomy” (ERTC) at first use.
Table 2: Consider including p-values for all variables, even if they are not statistically significant, to promote transparency.
References: Check the consistency of the formatting, especially the journal abbreviations
Author Response
Major comments:
Comment 1: Language and grammar: There are multiple grammatical issues throughout the manuscript that affect clarity. I recommend careful proofreading by a native English speaker or a professional editing service. For example: On line 20, 'their expression has been related...' should be 'has been related'. Moreover, inconsistent formatting of gene names needs to be fixed (e.g. KLF4, SOX2). Please follow standard gene/protein nomenclature (genes in italics and proteins in capital letters without italics).
Thank you for pointing this out. The genes have been formatted in italics and protein in capital letters. The manuscript has been revised for English by the expert editing Jiri Neuzil that contributed to the manuscript as writing review and editing.
Comment 2: RNA quality and sample integrity: The methods section should describe how RNA integrity was assessed beyond concentration and purity. Were samples of poor quality excluded? This is crucial for reproducibility and data robustness.
Thank you for pointing this out. The sentence ‘The RNA integrity was assessed by Qubit™ RNA IQ Assay Kits (Thermo Fisher Scientific)’ has been added to the ‘Quantitative RT-PCR’ paragraph (page 4, line 158-159)
Comment 3: Cut-off: The manuscript refers to 'high' and 'low' expression groups in the Kaplan–Meier analysis, but it is unclear how these were defined. Please clarify which statistical methods were used to determine the cut-offs.
We agree with this comment. Therefore, the cut-off has been clarified by adding in the statistical analysis paragraph the sentence ‘According to the median value, each pluripotent factor was categorized in low (below median value) and high (above median value) expression groups’ (page 5, line 200-202).
Comment 4: Lack of protein-level validation: all findings are based on mRNA levels. Given the complexity of post-transcriptional regulation, the absence of protein validation (e.g. by immunohistochemistry) undermines the translational value of the findings. While not mandatory, a brief discussion of this limitation is necessary.
We have accordingly done the Immunohistochemical (IHC) analysis for cMYC, SOX2 and OCT4. The tumor tissues were negative for SOX2, while positivity was found for OCT4 (25%) and cMYC (50%). The relationship between c-MYC and OCT4 expression and clinic-pathological parameters has been shown in Figure 6 (page 13, Survival analysis and association with clinicopathological parameters paragraph, line 388).
Comment 5: Survival and Prognostic Analysis: SOX2 is reported as an independent prognostic factor. Please clarify whether the multivariate analysis included all clinical variables (e.g. age, stage, smoking status and subtype) and specify the number of events per variable, to ensure the model is valid. Consider adding a forest plot to visualize hazard ratios.
We have settled the multivariate analysis by adding ITAC subtypes and type of surgery. First, we evaluated age, smoking, histological subtypes, tumor stage, and type of surgery in univariate analysis. Subsequently, significant predictors were included in multivariate analysis in association with pluripotent factors. In the multivariate survival analysis, SOX2 lost its significancy as prognostic factor, while c-MYC and OCT4 status in association with the type of surgery reached statistical significance. The data have been reported in Table 2, specifying the number of patients per variable. Forest plots were also included (page 10-12).
Comment 6: Contradictory interpretation of c-MYC: although higher c-MYC expression appears to be associated with better survival, the manuscript acknowledges that this is not statistically significant. This point is overemphasised in the discussion. It should be revised and toned down accordingly.
Thank you for pointing this out. Although c-MYC was not statistically significant in univariate analysis, it reached the significance in multivariate analysis. The IHC analysis confirmed the trend showing that higher c-MYC expression was associated with better survival. These results have been reported in Figure 6, and discussed (page 15, line 478).
We have accordingly deleted in the discussion the paragraph ‘c-MYC is an oncogenic factor involved in regulating a wide range of cellular processes, as well is a global regulator of chromatin structure through histone acetylation [41, 42]. It has a role in inducing apoptosis and cell differentiation, growth, and metabolism; it is also an oncogene that participates in processes of cell proliferation and inhibits differentiation, DNA replication, and metastasis [43]. Altogether, overexpression of c-MYC – or of its pa-ralogues, MYCN or MYCL – is a widespread event in most cancer types, and contributes to multiple hallmarks of the transformed phenotype, including cell intrinsic and systemic features, such as angiogenesis, modulation of the tumor microenvironment, or immune evasion [44]. Several observations also linked c-MYC to therapy resistance in solid tumors. In HER2-positive breast cancers, for example, amplification of the MYC locus identified a subgroup of patients with particularly poor prognosis when treated with adjuvant chem-otherapy [45]. Likewise, high expression of the MYC protein or a MYC-dependent gene signature predicted poor prognosis in patients suffering from estrogen receptor-positive breast cancer treated with adjuvant hormonal therapy [46]. In colon cancer, high expres-sion of the MYC transcript significantly correlated with tumor recurrence in patients who underwent adjuvant 5-fluorouracil (5FU) chemotherapy [47]. Finally, a retrospective analysis of several clinical studies suggested that elevated c-MYC expression might be associated with resistance to immune checkpoint inhibitor therapy in metastatic urotheli-al carcinoma [48]. c-MYC has been described as the main oncogenic factor related to tu-morigenesis and glycolysis as well as being linked to Wnt/β-catenin signaling in lung cancer [49]’.
Comment 7: The discussion would benefit from more detailed elaboration on the clinical implications and applications of these findings, especially the potential utility of SOX2 as a prognostic marker. Is there potential for risk stratification or treatment tailoring?
Thank you for pointing this out. However, by including in the multivariate analysis histological subtypes and type of surgery, the SOX 2 lost its statistical significance that was found in univariate analysis as reported in Table 2 and Forest plots (page 12, line 381).
Comment 8: Limitations: Please include a dedicated paragraph discussing the limitations of the study, such as the small sample size, single-centre design, lack of validation in independent cohorts and no protein-level confirmation.
We agree with this comment. As suggested, limitations of the study have been added in the conclusion paragraph (page 16, line 514).
Minor Comments:
Comment 9: Figures: Improve the contrast and font sizes of all figures. Legends should be fully self-explanatory.
We agree with this comment. Therefore, the figures have been improved in quality and size. As well more details have been given in the legend to figures.
Comment 10: Terminology: Consistently define and abbreviate “endonasal endoscopic resection” (ER) and “endonasal endoscopic resection with transnasal craniectomy” (ERTC) at first use.
We agree with this comment. Therefore, the endonasal endoscopic resection” (ER) and “endonasal endoscopic resection with transnasal craniectomy” (ERTC) have been defined at the first time and the abbreviations have been used along the text.
Comment 11: Table 2: Consider including p-values for all variables, even if they are not statistically significant, to promote transparency.
We agree with this comment. Therefore, Table 2 has been redone according to the wew results including the p value for all variables.
Comment 12: References: Check the consistency of the formatting, especially the journal abbreviations
As suggested, the references have been checked
Reviewer 2 Report
Comments and Suggestions for Authors
This study investigated the "stemness" phenotype of sinonasal intestinal-type adenocarcinoma (ITAC) in a cohort of 54 patients , with correlative outcomes.
The major conclusion was that SOX2 stands-alone as an independent prognostic factor for OS and DFS, speculating on the assumption that presence of stemness phenotype in ITAC may be associated with clinical response to treatment.
Several issues of concerns and critiques. Addressing these may enhance the quality and viability of this manuscript.
- No pathologist listed as co-author. How was the pathology addressed- review of ITACs, classification, morphology and phenotype, has to be detailed.
- The stemness phenotype was addressed by interrogating 4 transcription factors: KLF4, c_MYC, SOX2, OCT4 ("Yamanaka factors-include a proper reference for this), using qPCR. How where those 54 samples Q/C-ed for tumor content? Matching normal from same specimen/ patient or pooled?
- Results section-3.2 lists that "KLF4, c-MYC and SOX2 were lower expressed in tumor tissue compared with adjacent non-tumor tissue,; no changes for c_MYC and OCT4"- this is confusing and does not make sense with rest of the manuscript and discussion.
- The experimental designed is problematic- every scientific manuscript needs a validating methodology. Protein expression in FFPE samples is a readily available surrogate , at least for c-MYC, SOX2, OCT4- this needs to be done, in these samples. Document protein localization with respect to cellular composition of ITACs.
- To further address the stemness and CSC potential in this samples- ALDH expression should be investigated (flow and/ or qPCR, and IHC).
Comments on the Quality of English Language
The manuscript will benefit from scientific editing.
Author Response
Comment 1: No pathologist listed as co-author. How was the pathology addressed- review of ITACs, classification, morphology and phenotype, has to be detailed.
Thank you for pointing this out. The author Alessandra Filosa is the pathology at Polytechnic University of Marche Region, 60121 Ancona, Italy who performed IHC evaluation on ITAC tissues. Considering that the patients were recruited at the ENT Division of “Bellaria Hospital”–AUSL Bologna, the pathologist Maria P Foschini, who performed diagnosis of ITAC, including its classification and histological subtyping was added as author.
Comment 2: The stemness phenotype was addressed by interrogating 4 transcription factors: KLF4, c_MYC, SOX2, OCT4 ("Yamanaka factors-include a proper reference for this), using qPCR. How where those 54 samples Q/C-ed for tumor content? Matching normal from same specimen/ patient or pooled?
Thank you for pointing this out. The KLF4, cMYC, SOX2, OCT4 gene expression in tumour tissue was compared with its adjacent non-malignant counterpart as reported in ‘KLF4, c-MYC, SOX2, OCT4 and NANOG expression’ paragraph (page 8, line 270).
Comment 3: Results section-3.2 lists that "KLF4, c-MYC and SOX2 were lower expressed in tumor tissue compared with adjacent non-tumor tissue,; no changes for c-MYC and OCT4"- this is confusing and does not make sense with rest of the manuscript and discussion.
Thank you for pointing this out. "KLF4, SOX2 and NANOG were found down expressed in tumour respect to non-malignant tissue counterpart. However, within the tumour, high and low expression groups were identified according to the median value cut-off and then associated with patient outcomes.
Comment 4: The experimental designed is problematic- every scientific manuscript needs a validating methodology. Protein expression in FFPE samples is a readily available surrogate , at least for c-MYC, SOX2, OCT4- this needs to be done, in these samples. Document protein localization with respect to cellular composition of ITACs.
We agree with this comment. Therefore, protein expression of cMYC, SOX2 and OCT4 has been evaluated in FFPE samples by IHC, and a score expression (negative, weak, moderate strong) has been performed by a pathologist. The results are shown in Figure 6.
Commet 5: To further address the stemness and CSC potential in this samples- ALDH expression should be investigated (flow and/ or qPCR, and IHC).
We agree with this comment. Therefore, ALDH expression has been evaluated, and positivity was found only in 3 cases out of 17 patients. The data was not shown in the manuscript.
Comment 6: The manuscript will benefit from scientific editing.
Thank you for pointing this out. The manuscript has been revised for English by the expert editing Jiri Neuzil that contributed to the manuscript as writing review and editing.
Reviewer 3 Report
Comments and Suggestions for Authors
This is a welcome study on ITAC, a rare sinonasal tumor that is relatively under-studied.
Strong points include the even stage/subtype composition and the total number of tumors studied, all from patients treated in the same hospital in the same way. Also, published gene expression studies by RT-PCR on ITAC are rare, and the aim of this work is original. The correlation between SOX2 overexpression and survival is new and relevant to the literature.
However, there are also several comments to be made, given in order of the paper.
- Abstract: the conclusion is vague and response to treatment is not related to the results of this study.
- Introduction: is well-written.
- M+M: authors did not mention the collection of adjacent normal mucosa. Also it would be good to know at which distance from the tumor it was taken.
- M+M: The histopathological subtype of ITAC indicates the grade with papillary-colonic-solid types indicating well-moderate-poor differentiation, and mucinous type apart. Authors need to clarify/change this and adhere to the WHO classification. Also in table 1 and later tables and figures.
- M+M: the relative expression and fold-change in the tumor versus the normal mucosa should be better explained. Also, was each tumor compared with its own normal mucosa?
- It seems a low number of patients received adjuvant radiotherapy (9 of 54), perhaps this should be acknowledged or explained.
- Results: In figure 1 the expression of KLF4, SOX2 and NANOG in the normal mucosa shows a wide confidence interval. Is there an explanation? This should be discussed.
- Results: expression results are not correlated with ITAC subtype. As the genes under study affect stemness and differentiation, it would be very interesting to see how their expression is related to the histology of the tumors.
- Results: In all the literature, stage and ITAC subtype are strongly prognostic (statistically significant), so figure 4 should be changed to reflect this relation in the 54 tumors of this study, and it should be analyzed by log-rank.
- Results: in figure 5 is is not clear what is reflected by the red and black curves: the high or the low expression?
- Results: table 2 does not include ITAC subtype. This should be included.
- Discussion: Authors did not discuss previous publications on SOX2 in sinonasal tumors. Here are a few, there may well be more:
- Sex determining region Y-box 2 (SOX2) amplification is an independent indicator of disease recurrence in sinonasal cancer. Schröck A, Göke F, Wagner P, Bode M, Franzen A, Braun M, Huss S, Agaimy A, Ihrler S, Menon R, Kirsten R, Kristiansen G, Bootz F, Lengerke C, Perner S. PLoS One. 2013;8(3):e59201.
- SOX2 expression in hypopharyngeal, laryngeal, and sinonasal squamous cell carcinoma. González-Márquez R, Llorente JL, Rodrigo JP, García-Pedrero JM, Álvarez-Marcos C, Suárez C, Hermsen MA. Hum Pathol. 2014 Apr;45(4):851-7. doi: 10.1016/j.humpath.2013.12.004.
- Wang, X.; Liang, Y.; Chen, Q.; Xu, H.-M.; Ge, N.; Luo, R.-Z.; Shao, J.-Y.; He, Z.; Zeng, Y.-X.; Kang, T.; et al. Prognostic Significance of SOX2 Expression in Nasopharyngeal Carcinoma. Cancer Investig. 2012, 30, 79–85.
- Sox2 and betaIII-Tubulin as Biomarkers of Drug Resistance in Poorly Differentiated Sinonasal Carcinomas. López L, Fernández-Vañes L, Cabal VN, García-Marín R, Suárez-Fernández L, Codina-Martínez H, Lorenzo-Guerra SL, Vivanco B, Blanco-Lorenzo V, Llorente JL, López F, Hermsen MA. J Pers Med. 2023 Oct 18;13(10):1504.
- Discussion: Related to comment 12, the authors discuss KLF4 and SOX2 literature on oral SCC, but ITAC an adenocarcinoma, a very different tumor type from squamous cell carcinoma. When looking outside the sinonasal area, it may be more interesting to discuss literature on adenocarcinoma in other organs than on squamous cell carcinoma in other organs.
- The list of 61 references contain 9 of the authors, that seems a lot. Of these 9, two concern ITAC, and one sinonasal papilloma, but six are studies on larynx SCC and don't seem relevant the the present subject of study.
- References nr 23 and 30 are the same.
Comments on the Quality of English Language
Overall ok, but some sentences are awkward.
Author Response
Comment 1: Abstract: the conclusion is vague and response to treatment is not related to the results of this study.
Thank you for pointing this out. We have accordingly rewritten the conclusion
Comment 2: Introduction: is well-written.
Thank you for the appreciation
Comment 3: M+M: authors did not mention the collection of adjacent normal mucosa. Also it would be good to know at which distance from the tumor it was taken.
We agree with this comment. Therefore, the sentence ‘Tumor tissue and its adjacent non-malignant tissue were collected and stored at -80°C has been added in M+M, ‘Patients and tissue specimens’ paragraph (page 4, line 148).
Commet 4: M+M: The histopathological subtype of ITAC indicates the grade with papillary-colonic-solid types indicating well-moderate-poor differentiation, and mucinous type apart. Authors need to clarify/change this and adhere to the WHO classification. Also in table 1 and later tables and figures.
We agree with this comment. Therefore, a paragraph ‘According to the WHO classification (15), ITACs were classified into histological subtypes indicating well-moderate-poor differentiation, typical for the papillary, colonic, and solid subtypes. The mucinous type displays mucin-filled glands, or cell clusters’ has been added at ‘Patients and tissue specimens’ paragraph (page 4, line 141).
Commet 5: M+M: the relative expression and fold-change in the tumor versus the normal mucosa should be better explained. Also, was each tumor compared with its own normal mucosa?
Thank you for pointing this out. The levels of KLF4, c-MYC, SOX2, OCT4 and NANOG were evaluated in tumor and its adjacent non-malignant counterpart and expressed as a relative expression using GAPDH as housekeeping gene (2-ΔCT) as reported in the ‘Quantitative RT-PCR’ paragraph (page 4-5, line 172). The results were not expressed as fold-change, tumor compared with its own normal mucosa (2-ΔΔCT).
Comment 6: It seems a low number of patients received adjuvant radiotherapy (9 of 54), perhaps this should be acknowledged or explained.
We agree with this comment. The low number of patients who received adjuvant radiotherapy could be explained by the fact that they were not scheduled for RT due to different reasons, such as tumor stage, surgical approach, tumor invasiveness, patient age.
Commet 7: Results: In figure 1 the expression of KLF4, SOX2 and NANOG in the normal mucosa shows a wide confidence interval. Is there an explanation? This should be discussed.
We agree with this comment. An explanation should be that tumor, which is close to the non-malignant counterpart may affect the expression of pluripotent factors in a paracrine manner. The sentence ‘The high variability in the expression of pluripotent factors found in the adjacent non-malignant counterpart may be due to the paracrine effect of tumors’ has been added in the discussion (page 14, line 410).
Comment 8: Results: expression results are not correlated with ITAC subtype. As the genes under study affect stemness and differentiation, it would be very interesting to see how their expression is related to the histology of the tumors.
Thank you for pointing this out. We have accordingly evaluated the distribution of pluripotent factors among ITAC subtypes. No significant differences in pluripotent gene expression among histotypes have been found.
Comment 9: Results: In all the literature, stage and ITAC subtype are strongly prognostic (statistically significant), so figure 4 should be changed to reflect this relation in the 54 tumors of this study, and it should be analyzed by log-rank.
We agree with this point. We have accordingly deleted the figure 4, and the 5-year OS of ITAC subtypes has been reported in the ‘Survival analysis and association with clinicopathological parameters’ paragraph (page 10, line 319). Moreover, ITAC subtypes have been included both in univariate and multivariate analysis.
Comment 10: Results: in figure 5 is is not clear what is reflected by the red and black curves: the high or the low expression?
We agree with this point. The high and low groups refer to the high and low expression of pluripotent factors within the tumor using the median value as cut-off as reported in ‘Survival analysis and association with clinicopathological parameters’ paragraph (page 10, line 323)
Comment 11: Results: table 2 does not include ITAC subtype. This should be included.
We agree with this point. Therefore, Table 2 has been rebuilt by including ITAC subtype and type of surgical, specifying the number of patients per variable. Forest plots have been also added (page 10-12).
Comment 12: Discussion: Authors did not discuss previous publications on SOX2 in sinonasal tumors. Here are a few, there may well be more:
Sex determining region Y-box 2 (SOX2) amplification is an independent indicator of disease recurrence in sinonasal cancer. Schröck A, Göke F, Wagner P, Bode M, Franzen A, Braun M, Huss S, Agaimy A, Ihrler S, Menon R, Kirsten R, Kristiansen G, Bootz F, Lengerke C, Perner S. PLoS One. 2013;8(3):e59201.
SOX2 expression in hypopharyngeal, laryngeal, and sinonasal squamous cell carcinoma. González-Márquez R, Llorente JL, Rodrigo JP, García-Pedrero JM, Álvarez-Marcos C, Suárez C, Hermsen MA. Hum Pathol. 2014 Apr;45(4):851-7. doi: 10.1016/j.humpath.2013.12.004.
Wang, X.; Liang, Y.; Chen, Q.; Xu, H.-M.; Ge, N.; Luo, R.-Z.; Shao, J.-Y.; He, Z.; Zeng, Y.-X.; Kang, T.; et al. Prognostic Significance of SOX2 Expression in Nasopharyngeal Carcinoma. Cancer Investig. 2012, 30, 79–85.
Sox2 and betaIII-Tubulin as Biomarkers of Drug Resistance in Poorly Differentiated Sinonasal Carcinomas. López L, Fernández-Vañes L, Cabal VN, García-Marín R, Suárez-Fernández L, Codina-Martínez H, Lorenzo-Guerra SL, Vivanco B, Blanco-Lorenzo V, Llorente JL, López F, Hermsen MA. J Pers Med. 2023 Oct 18;13(10):1504.
Thank you for pointing this out. However, In the multivariate survival analysis, SOX2 lost its significancy as prognostic factor, while c-MYC and OCT4 status in association with the type of surgery reached statistical significance. Therefore, the prognostic value of SOX2 was not discussed further.
Comment 13: Discussion: Related to comment 12, the authors discuss KLF4 and SOX2 literature on oral SCC, but ITAC an adenocarcinoma, a very different tumor type from squamous cell carcinoma. When looking outside the sinonasal area, it may be more interesting to discuss literature on adenocarcinoma in other organs than on squamous cell carcinoma in other organs.
Thank you for pointing this out. The discussion has been changed in accordance with new results.
Comment 14: The list of 61 references contain 9 of the authors, that seems a lot. Of these 9, two concern ITAC, and one sinonasal papilloma, but six are studies on larynx SCC and don't seem relevant the the present subject of study.
We agree with this comment. The references have been updated in accordance with new results.
Comment 15: References nr 23 and 30 are the same.
Thank you for pointing this out. The references have been corrected
Comment 16: Overall ok, but some sentences are awkward.
The English has been revised by the expert editing Jiri Neuzil that contributed to the manuscript as writing review and editing.
Round 2
Reviewer 2 Report
Comments and Suggestions for Authors
Illustrations will benefit by show H&E examples of ITAC- with c-myc and OCT4 IHCs.
Also, Sox2 is a doable IHC- please perform, at least on a subset of samples- as correlative findings.
Author Response
The manuscript has been changed to better clarify the study design, results and conclusions. The aim has been better elucidated (page 4, line 133). As well, the ‘Quantitative RT-PCR’ method has been better described (page 4, line 187). The paragraph (page 10, line 345) ‘The study was conducted over a median (±SE) follow-up period of 42 ± 5 months [95% CI 32-52] and survival time ranging from 2.6 to 231 months. 5-year OS was 65.3 %, varying according to the histological subtype, with 80 % for well differentiated papillary subtype, 50 % for the colonic subtype, 57 % for the solid subtype, and 50 % for the mucinous subtype’ has been moved in the result ‘Study population’ paragraph (page 6, line 257).
Further, in the result the paragraphs ‘KLF4, c-MYC, SOX2, OCT4 and NANOG expression’ (page 8, line 309) and ‘Survival analysis and association with clinicopathological parameters’ (page 10, line 366) have been described in detail. To better understand the stemness role in ITAC, the ADLH1A1 factor has been also evaluated in IHC and reported in Figure 6. As suggested, the figure and tables have been better presented.
Comment 1: Illustrations will benefit by show H&E examples of ITAC- with c-myc and OCT4 IHCs.
Thank you for pointing out this comment. As suggested, H&E images have been added to the IHC staining in Figure 5 and Figure 6.
Comment 2: Also, Sox2 is a doable IHC- please perform, at least on a subset of samples- as correlative findings.
We agree with this comment. Therefore, SOX2 has been evaluated in patient sub-population and showed in Figure 5. To clarify the stemness prognostic role in ITAC, the ALDH1A1 tumor positivity was also detected in IHC. As mentioned on page 13 line 451, no SOX2 positivity was detected, while positivity to ALDH1A1 was found in 9 of 17 cases (52%), and its expression was associated with a favorable prognosis (page 14, line 474). The role of ALDH1A1 has been also discussed on page 17, line 604.
Reviewer 3 Report
Comments and Suggestions for Authors
The authors have carefully addressed all comments, and I have no more remarks.
Author Response
Comment 1: The authors have carefully addressed all comments, and I have no more remarks.
Thank you for revising the manuscript.